# Tribological Characterization of a Novel Ceramic–Epoxy–Kevlar Composite

**DOI:** 10.3390/polym16060785

**Published:** 2024-03-12

**Authors:** Yassin Fouad, Abdulrahman A. Aleid, Omer Osman, Necar Merah, Amjad Shaarawi, Ali Hijles, Fawzia Waluyo

**Affiliations:** 1Department of Mechanical Engineering, King Fahd University of Petroleum and Minerals, Dhahran 31261, Saudi Arabia; g202213740@kfupm.edu.sa (Y.F.); rahman.aleid@gmail.com (A.A.A.);; 2Interdisciplinary Research Center for Advanced Materials, King Fahd University of Petroleum and Minerals, Dhahran 31261, Saudi Arabia; 3Drilling Technology Team, EXPEC Advanced Research Center, Saudi Aramco, Dhahran 31261, Saudi Arabiaali.hijles@aramco.com (A.H.);

**Keywords:** casing wear, wear factor, drillpipe tool joint, ceramic, kevlar, epoxy, wear mechanisms

## Abstract

This work aims to explore the effect of side load and rotational speed on the tribological behavior of a novel ceramic–epoxy composite in Kevlar matrix casing lining that is in contact with a rotating drillpipe tool joint (DP-TJ) coated with the same composite. Three rotational speeds (65, 115, and 154 rpm) and three side loads (500, 700, and 1000 N) were considered under water-based mud (WBM) lubrication. Wear depths, volumes, and specific casing wear rates (K) were determined for each combination of speed and load. The wear depth and K were found to increase with an increasing applied side load. However, the specific casing wear rate at the rotational speed of 115 rpm was found to be the lowest among the three speeds. This is mainly due to a probable lubrication regime change from boundary lubrication at 65 rpm to hydrodynamic lubrication with a thick lubricant film at 115 rpm. The digital microscope images were used to determine the wear mechanism, showing that at low speeds, the main mechanism was abrasive wear, but the increase in the speed brought about more adhesive wear. In contrast, the change in the side load does not affect the wear mechanism of the casing. Scanning electron microscopy and energy-dispersive spectroscopy (EDS) were used to analyze the surface and composition of the novel material before and after the wear tests.

## 1. Introduction

During oil well drilling operations, numerous factors can cause the casing to fail, leading to catastrophic financial and environmental results. Casing failure is a complex problem caused by various factors that include, but are not limited to, corrosion, wear, faulty design, and the surrounding environment [1,2,3]. Casing wear is mainly caused by the rotation of the drillpipe tool joint (DP-TJ) against the inner surface of the casing [4].

In general, wear can be categorized based on the fundamental wear mechanisms involved, including adhesive wear, abrasive wear, corrosive wear, and surface fatigue wear [1,5,6,7]. Corrosion poses a significant problem when using metallic casings [8,9]. The use of metallic parts in oil-lubricated systems can lead to significant tribocorrosion challenges, depending on the operating environment, and can be costly [5,10,11].

In the last two decades, there has been a substantial increase in the use of non-metallic materials in applications for harsh environments and wear resistance [12]. This is attributable to elements such as the creation of unique polymer-based composite materials like Kevlar and the incorporation of hardening lubricating elements like ceramics, graphite, and PTFE. This offered excellent resistance to impact loads and decreased friction and wear [13,14,15]. The tribological behavior of composite materials cannot be exactly described by a precise model due to the various factors influencing wear, including the hardening ceramic concentration, counter face, fiber orientation, sliding speed, lubrication, and temperature [16,17].

Kevlar fiber-reinforced polymer composite, famous for its exceptional tensile strength-to-weight ratio, which is around five times higher than the value of steel, is often combined with carbon and glass fibers to enhance reinforcing properties [18,19]. Kevlar fabric shows a tensile strength of 3620 MPa and a very low relative density of 1.44 [19]. Liu et al. [20] found that the Kevlar-pulp concentration improved wear resistance. They also found that the presence of water provided boundary lubrication that enhanced wear resistance and cooling of the composite. Interestingly, the tribological behavior of some Kevlar-based composites proved to be affected by the oscillating frequency-bearing load.

Extensive research has been carried out to explore the sliding wear characteristics of advanced ceramics [21,22]. Studies constantly show that tribofilms, thin layers of fine wear debris particles, influence ceramic sliding wear behavior. These tribofilms form at contact interfaces, highlighting their importance in understanding wear mechanisms and their impact on friction and wear [21]. Denape and Lamon [22] provided proof that both the wear rate and the friction response are controlled by the movement of wear debris within the sliding interface. They found that the quantity of wear debris that becomes trapped inside the system has a substantial impact on the wear characteristics of the ceramics under examination.

In their research, Kurahatti et al. [23,24] observed the improvement caused by the addition of nano-zirconia particles on the tribological properties. The authors [23] utilized a dry pin-on-disk test and observed the tribological properties of nano-zirconia (ZrO_2_) bismaleimide (BMI) composites sliding against the En-32 steel disk. They highlighted the significant reduction in the wear rate and friction, compared to pure BMI, and observed a 78% decrease in the specific wear rate obtained at 5 wt.% of zirconia filler. The authors also highlighted the correlation between the hardness and the wear performance. In their later study [24], they found that, compared to the unfilled epoxy, a 95% reduction in the wear rate was obtained when adding 0.5 wt.% of zirconia. The resulting wear mechanism was observed to be mild abrasion. Wang et al. [25] concluded that the addition of 0.5 wt.% of the ZrO_2_ nanohybrid to phenolic resin improved the wear rate and the coefficient of friction by 30.6% and 21.8%, respectively. Yi et al. [26] found that the addition of ZrO_2_ to Ti_3_C_2_ epoxy composites yielded a lower wear rate with a higher coefficient of friction. The impact of the nano-SiC particle volume fraction on tribological behavior in Kevlar/E Glass/Epoxy composite was studied by Sudarshana et al. [27] in dry sliding conditions. As the counter face, pin-on-disk tests were performed using ASTM G-99 steel. According to the results, the composite’s wear rate lowers as the volume fraction of SiC rises. By firmly connecting the matrix and the fibers, SiC particles were found to lower the wear rate by preventing material deterioration due to adhesion.

Based on the above-mentioned advantages, fiber-reinforced epoxy–ceramic composites have the potential to be used as coatings for metallic and non-metallic casings and DP-TJ, especially in corrosive environments. The above studies have addressed the tribological behavior of a number of polymer-reinforced composites and nanocomposites. The effects of zirconia and other particles on the wear and friction properties of polymer-based composites are usually determined using pin-on-disk. To the authors’ knowledge, none of the studies have addressed the tribological characteristics of the present epoxy–Kevlar–ceramic composite. Furthermore, none of the studies have used real oil/gas drilling casings and drillpipe joints with composite linings. In the present work, a modified and automated lathe machine with a data acquisition system was used to perform various wear tests on a real casing pipe with a Kevlar-reinforced ceramic composite lining in contact with real DP-TJ coated with the same material. Three DP-TJ rotational speeds (65, 115, and 154 rpm) and three contact loads (500, 700, and 1000 N) were considered under water-based mud (WBM) lubrication. This paper aims to study the tribological properties of the novel material in terms of wear depths, volumes, and specific casing wear rates (K) under conditions similar to oil well drilling operations. The wear mechanisms under different testing conditions will also be investigated.

## 2. Material and Methods

### 2.1. Materials and Equipment

The casing samples comprise P110 carbon steel casing coated with a proprietary protective layer of Kevlar–epoxy composite filled with a 2.5% volume fraction of zirconia particles. Sixty-degree arc-shaped samples were cut off from the as-received casing pipes. Real photos of the casing samples are shown in Figure 1. The casing sample has a width of 35 mm and a total thickness of 17 mm, of which 3 to 5 mm is a composite coating. The drill-pipe tool-joint (DP-TJ) (shown in Figure 1c,d) is made of carbon steel, coated with three distinct protective layers, with the outermost layer consisting of Kevlar–ceramic–epoxy composite. The dimensions of the as-received casing samples and DP-TJ are summarized in Table 1. The tool joint was machined down to an outer diameter of 137.2 mm to eliminate the irregularities in the outer surface. A 4 mm diameter hole was drilled in the steel backing of the casing sample to accommodate the insertion of a temperature sensor, which is essential for monitoring the thermal response of the samples during the wear tests.

The chemical composition of the composite casing was provided by the manufacturer as Cycloaliphatic Amine/Novolac Epoxy with nanoparticulate-modified epoxy additive. Also, it included surface-treated ZrO_2_ beads of various dimensions with a proprietary filler system, all of which were poured inside an open honeycomb Kevlar matrix (Figure 1e). Table 2 below illustrates the mechanical properties as provided by the supplier.

The wear tests were performed using the experimental setup, consisting of a modified late machine developed in earlier work [28,29,30,31], wherein the conventional cutting tool was replaced with a custom test specimen holder. Additionally, the tool joint was securely installed into the rotating spindle of the machine to simulate rotation; schematics of the setup are shown in Figure 2.

Testing was performed under water-based mud (WBM) lubrication, whereas DP-TJ partially bathes in the mud. Table 3 illustrates the formulation of the WBM used in the tests. The load was applied using a Nema 34 closed-loop stepper motor with a HBS860H driver. The casing holder was rigidly mounted on a dynamometer type 9139AA (Kistler Instrument Corp., Hudson, NY, USA). The dynamometer measures the three-dimensional loads applied to the casing during the wear tests. A waterproof DS18B20 digital thermal probe sensor (Milsight, Xiamen, China) was used to measure the average casing temperature during the wear tests. Moreover, an electronic digital micron indicator, with a least count of 1 μm and accuracy of ±4 μm, was used to measure the radial displacement of the casing specimen (maximum wear depth). The reliability of the depth measuring system has been validated by Osman et al. [29], who compared the wear depth measured by the digital indicator to the one obtained by the 3D profilometer. The dynamometer, digital thermal probe, digital micron indicator, and stepper motor are all connected to a microcontroller that controls the applied side loads and collects the data. Through a feedback loop based on the reading from the dynamometer, the programmed Arduino board controls the stepper motor to keep the applied load within the desired range.

### 2.2. Experimental Procedure

Three rotational speeds (65, 115, and 154 rpm) and three side loads (500, 700, and 1000 N) were used under WBM lubrication. The DP-TJ speeds are close to the real drillstring speeds of 100 to 150 rpm and the side loads are selected such that the unit loads would be within the field condition pressures of 0.3 to 0.5 MPa [31]. Each casing specimen was placed in direct contact with the rotating DP-TJ under the selected applied side load and speed for 5 h. The collected data concerning the wear depth, applied load, and rotational speed were subsequently used to calculate the wear volume, specific casing wear rates, coefficient of fraction, and other needed parameters. Twelve wear tests were performed under different rotational speeds and side loads, as shown in Table 4.

### 2.3. Characterization

The hardness of the Kevlar–ceramic protective layer was measured before and after testing according to ASTM D2240 [32], using the Shore D hardness tester of Digi Test (BAREISS PRÜFGERÄTEBAU GMBH, Kolbeweg, Germany). The digital microscopic images were taken using the DSX510 (Olympus IMS, Waltham, MA, USA) to characterize the composite lining before and after testing at different locations and angles. A scanning electron microscope (SEM) (JEOL SEM model JSM-6460 with an EDS facility and JEOL Gold Sputter model JFC-1100, Tokyo, Japan) was used to analyze the worn surfaces and study the chemical composition and elemental distribution on the surface before and after the wear tests.

## 3. Results and Discussion

### 3.1. Hardness Evaluation

The average shore D hardness (SHRD) values were obtained for a selected number of samples before and after the wear test. An average of nine readings, taken at different points, were recorded. The average shore hardness value of the as-received samples was found to be 87.8 ± 2.3. This value is very close to that reported by the manufacturer.

The effect of the testing conditions on the hardness of the epoxy–Kevlar composite lining was also investigated. Figure 3 shows how the hardness values of the composite layer vary with a side load at 115 rpm. It is observed that, in general, the hardness values slightly decrease after the test due to the removal of the first protective layer. For the tested samples, the hardness value increases with the applied load. On the other hand, the variation in the hardness values for specimens tested under 500 N and three speeds is shown in Figure 4. It is noticed that the hardness value slightly reduces as the speed increases to 115 rpm.

Figure 5 shows a micrograph of the surface of the as-received casing. The Kevlar fibers are found to form a honeycomb structure. This structure affects the value of the measured hardness. It is observed that the highest hardness values are located at the junctions of the open Kevlar matrix, while the minimum values are located at the center of the hexagonal shape. After the wear test (Figure 6) these locations change, as the highest value is located at the center of the hexagonal, and the minimum value is at the junctions connecting different hexagons. This is mainly due to the presence of uncovered, very hard zirconia particles. The detailed microscopic analyses of Section 3.4 reveal a strong presence of zirconia beads on the worn surfaces of the lining at all combinations of speeds and loads.

### 3.2. Wear Depth and Wear Volume

The graphs in Figure 7 and Figure 8 show the change in the measured wear depth with testing time. Figure 7 displays representative wear depth data that were recorded over time during the casing wear test at 65 rpm and at various side loads (500, 700, and 1000 N). It is observed that the maximum wear depth for 500 and 700 N varies linearly with the testing time, resulting in an approximately constant wear rate, which increases with an applied load. However, for the side load of 1000 N, the wear rate increases drastically until most of the composite coating wears out at around 90 min, after which the wear rate decreases. Figure 8 shows how the rotational speed affects the wear depth. It is noticed that, in general, the wear rate increases as the speed increases. This is mainly due to the increase in the total sliding distance, which is proportional to the rotational speed. As will be discussed in a later section, the shift in the lubrication regime from boundary to hydrodynamic lubrication is most likely the cause of the observed decreased wear rate at 115 rpm.

The wear volume of the tested composite lining of casing samples was calculated based on the measured maximum wear depth values obtained at the end of the test after deducing the change in DP-TJ radius. Figure 9 represents the crescent-shaped wear groove of the casing, while Equation (1) is used to estimate the wear volume per unit specimen width (WV) (mm^3^/mm).
(1)$$\mathrm{WV}=\left({\beta r}^{2}+2\sqrt{Q\left(Q-R\right)\left(Q-S\right)\left(Q-r\right)}-{\alpha R}^{2}\right)$$
with
$$Q=\frac{R+r+S}{2},$$
S=R−(r−w),
$$\cos\alpha =\left(\frac{R^{2}+S^{2}-r^{2}}{2\mathrm{RS}}\right),$$
$$\beta =\mathrm{arctg}\left(\frac{R\sin\alpha }{R\cos\alpha -S}\right)$$
where

w is the maximum depth of the wear (mm);

R is the inner radius of the casing (mm);

r is the outer radius of the drill pipe (mm).

**Figure 9 polymers-16-00785-f009:**
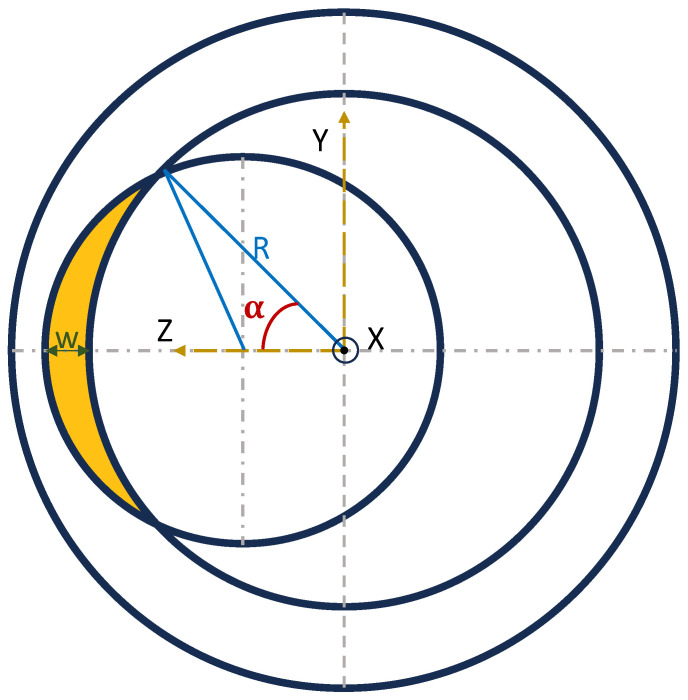
Crescent-shaped wear groove. [31].

### 3.3. Specific Wear Rates

The specific wear rate, also known as the casing wear factor, (K) in MPa^−1^ was calculated for the composite using the following equation:(2)K=V/PL
with
L=Π×N×t×D
V=WV×Sample width
where

V: wear volume (mm^3^);

P: radial load (N);

L: total sliding distance (mm);

N: rotational speed (rpm);

t: testing time (min);

r: tool joint diameter (mm).

The calculated specific wear rate (K) values for the Kevlar–epoxy–ceramic composite casing lining under different side loads and rotational speeds are illustrated in Table 5.

Figure 10 shows a 3D illustration of how the specific wear rate changes with different loads and speeds. It is clear that K increases with the applied side load. This is due to the increase in contact pressure, which causes grooves, scratches, or removal of material by two-body abrasion, as will be demonstrated in the next section. Moreover, it can be observed that changing the load from 500 N to 700 N and from 700 N to 1000 N at the rotational speed of 115 rpm results in more than double the specific wear rate value. The effect of the contact load, between 500 and 700 N, becomes more important at 65 rpm (nearly 10 times increase in K), while at 154 rpm, the increase is about 27%. Finally, the increase of the load to 1000 N leads to a drastic jump in the specific wear rate at 65 rpm and 154 rpm.

The results of Figure 10 and Table 5 reveal that the average value of K is the lowest at 115 rpm for all three levels of contact loads. At the speed of 65 rpm, there is probably a boundary lubrication with a thin lubricant film that has frequent solid-to-solid contact, which means that zirconia plays a role as abrasive particles at a lower speed, indicated by the high average coefficient of friction when compared to the one at the speed of 115 rpm, as shown in Figure 11.

The reduction in the specific wear rate and coefficient of friction that occurs at the speed of 115 rpm can be explained by exploring two possible reasons. First, the lubrication regime probably changes from boundary lubrication at 65 rpm to hydrodynamic lubrication with a thick lubricant film at 115 rpm. The second reason could be explained by the variation in the coefficient of friction (COF) presented in Figure 12. Figure 12b shows repeated drops in the COF, which is probably due to the “Solid-Liquid Slip” effect. This effect has been explored by many researchers [33,34,35,36,37]. Spikes et al. [33] mentioned that the presence of a thin lubricant layer at the interface of a solid that does not adhere to the solid surface causes what is known as boundary slip. This behavior occurs in high-shear-rate environments that affect the flow behavior of the fluid near the surface. Vinogradova [35] explained that the hydrophobicity of contacting surfaces could affect the viscosity near the wall, which causes the slip behavior, leading to a drop in COF.

When the speed is increased from 115 rpm to 154 rpm, the thickness of the fluid film increases, leading to higher viscous forces. This behavior is similar to that of the journal bearing with the characteristics number (µN/P), where P is the unit load, N is the rotating speed, and µ is the viscosity of the lubricant. An increase in the N results in an increase in the bearing characteristic number, which, according to the McKee brothers’ curve, will result in a thicker lubricant film and higher friction coefficient. It is to be mentioned that the lubrication system in the present tests is more complicated than that of an oil-lubricated journal bearing. The film is a mixture of solids such as zirconia and silica particles with water-based mud.

The fluid film plays a role in shifting the cause of wear from solid-to-solid contact to material pullout caused by the viscous shear forces of the hydrodynamic lubrication. These higher forces help overcome the slip effect at the solid surface, leading to the pullout of small chunks of epoxy along with zirconia particles. Some of the removed hard zirconia particles could potentially play the role of an abrasive third body, leading to an increase in the COF. Finally, the effect of the fluid film could be noticed in the drop in the coefficient of friction that happens at 1000 N, which could be related to the squeeze effect [38].

### 3.4. Wear Mechanisms

An optical microscope and scanning electron microscope (SEM) were used to study the effects of the applied side load and rotational speed on the wear characteristics of the casing lining surface before and after the wear tests.

The microscopic image of Figure 13 shows a fish scale-like structure of the surface of the first layer of the as-received sample. As will be seen later, this layer is mainly composed of epoxy and silica. Figure 14 shows the microstructure of the worn lining at different DP-TJ rotational speeds and side loads. Figure 14 a reveals that, at a low speed and low applied side load, abrasive wear is the dominant mechanism. As the speed increases from 65 to 115 rpm, more adhesive wear is observed (Figure 14b). The latter is characterized by the removal of epoxy chunks along with zirconia particles. Some of the fallen spherical zirconia beads, combined with hydrodynamic lubrication, would act as rollers, leading to a decrease in the coefficient of friction observed at 115 rpm. The effect of the side load on the wear mechanism is less pronounced at 115 rpm, as can be seen in Figure 14b,c. Both abrasive and adhesive wear mechanisms with zirconia removal are present for the two loads of 500 and 700 N. The SEM micrograph of Figure 15 confirms the presence of both abrasive and adhesive wear mechanisms. It also shows that there are two types of holes, one being the large holes caused by the adhesive wear while the other is due to the removal of the zirconia beads during the sliding process. This phenomenon has also been observed by Che et al. [39], who tested graphene oxide sheets–zirconia spheres (ZrO_2_-rGO) nanohybrids in dry conditions.

Figure 16 is an SEM micrograph of the as-received specimen, and Table 6 is the EDS analysis of the surface, which shows that the upper layer is mainly epoxy with silicon and some aluminum. Figure 17 also shows a high count-per-second per electron Volt (cps/eV) for the carbon content, which is around 3 cps/eV. The presence of silicate in epoxy is probably due to the addition of nanosilicates to improve the stiffness and strength of the composite and reduce its moisture uptake. The SEM micrograph in Figure 18 and the EDS results in Table 7 of the surface of a sample tested under 500 N and 154 rpm show that the morphology of the worn surface is completely different from that of the untested surface, and the EDS analyses of different spectra illustrate a varied chemical composition. After removing the first coating layer, the zirconia beads located below the surface appear in spectra 1, 2, 3, and 6. The size of the Zr particles is seen to vary between 50 and 80 microns. Spectra 4 and 5 are void of Zr and are mainly epoxy resin.

## 4. Conclusions

A modified and automated lathe machine was employed to perform wear tests on real casing pipes with Kevlar–epoxy–ceramic composite lining, using a field-size DP-TJ coated with the same composite as the counterface and real drilling water-based mud. The tests aimed to characterize the tribological properties and behavior of the novel material for potential use as coatings for casings and the DP-TJ in oil and gas well drilling. The main findings are summarized as follows:The specific wear rate increases with contact load because of the increase in contact pressure and the presence of more asperities.The specific wear rate was significantly affected by the rotation speed with the minimum wear at the intermediate speed of 115 rpm. The primary cause of this behavior is the probable change in the lubrication regime from thin to thick film. The presence of zirconia tribofilms may also be contributing to the observed improvement in K.The abrasive wear mechanism was found to be dominant at lower speeds and loads, while both abrasive and adhesive wear were present at higher speeds. The effect of side load on the wear mechanisms is not very important.The EDS analyses showed that the material is composed of two different layers, where the surface is mostly aluminum and silica mixed with an epoxy matrix. The predominant elements in the lower layers are a mix of alumina, zirconia, and silica embedded in an epoxy matrix.

## Figures and Tables

**Figure 1 polymers-16-00785-f001:**
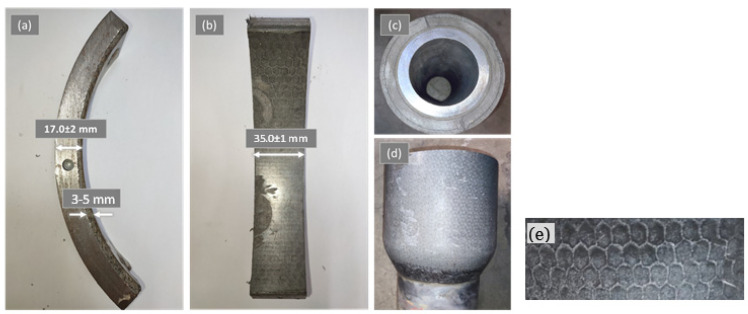
Real photos of the received samples where (**a**,**b**) show the casing, (**c**,**d**) the drill-pipe tool-joint, and (**e**) a closeup of the honeycomb structure.

**Figure 2 polymers-16-00785-f002:**
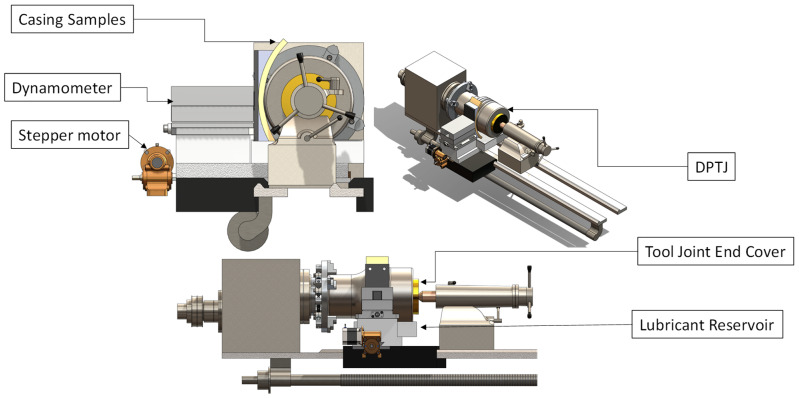
Three-dimensional model of the casing wear testing facility.

**Figure 3 polymers-16-00785-f003:**
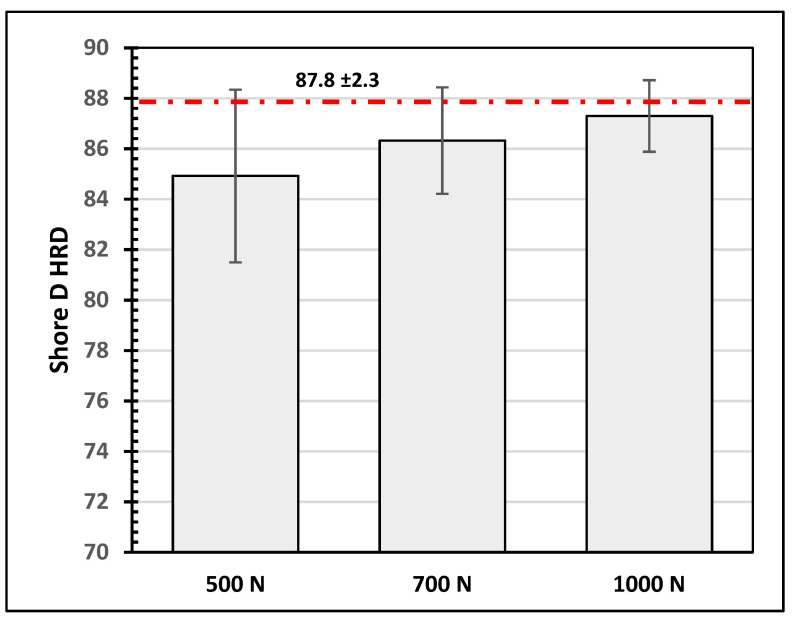
Shore hardness values for samples tested at 115 rpm and three different loads.

**Figure 4 polymers-16-00785-f004:**
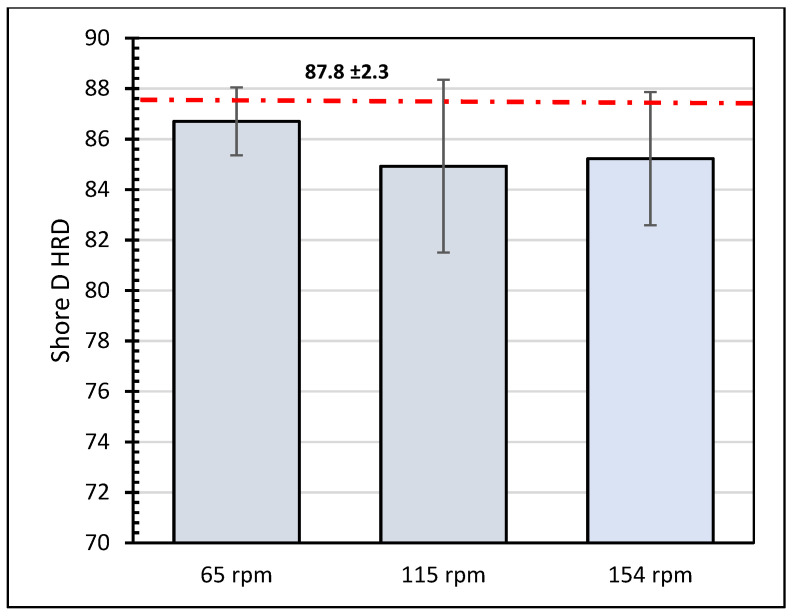
Shore hardness values for samples tested under 500 N and three different speeds.

**Figure 5 polymers-16-00785-f005:**
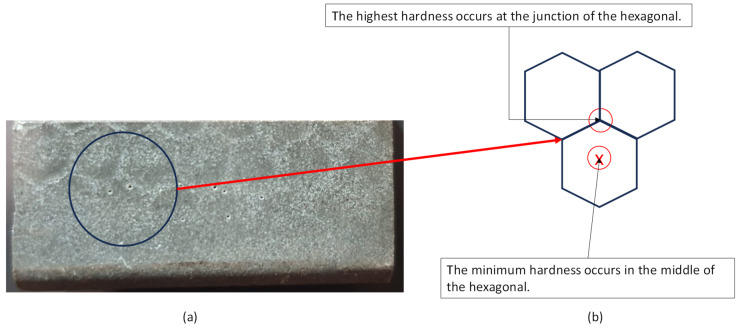
(**a**) Untested casing sample; (**b**) schematic showing the variation in hardness at different locations.

**Figure 6 polymers-16-00785-f006:**
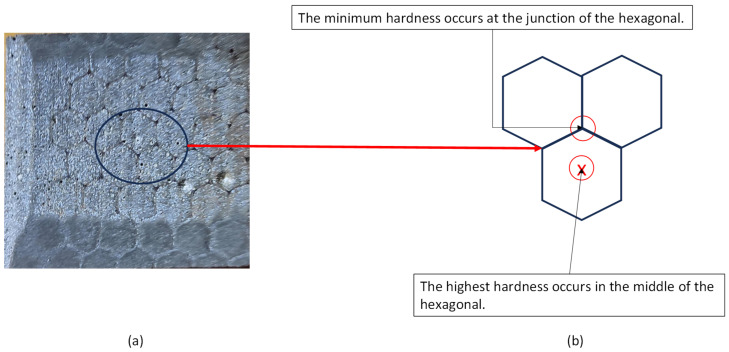
(**a**) Casing sample tested under 500 N and 115 rpm; (**b**) schematic showing the variation in hardness at different locations.

**Figure 7 polymers-16-00785-f007:**
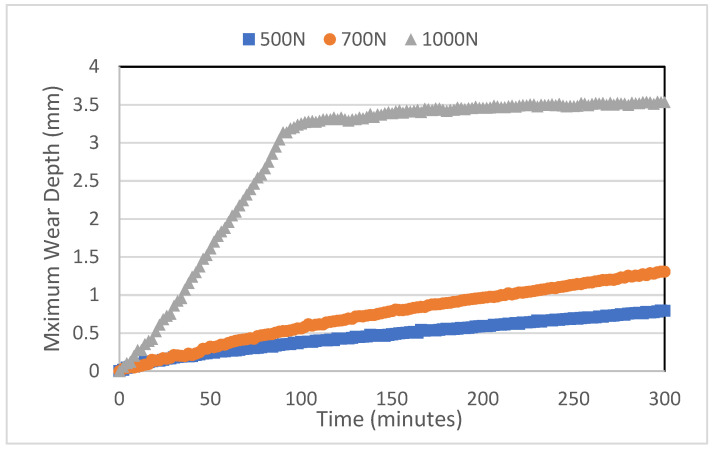
The maximum wear depth tested at 154 rpm and the different loads (500 N, 700 N, and 1000 N) versus time (minutes).

**Figure 8 polymers-16-00785-f008:**
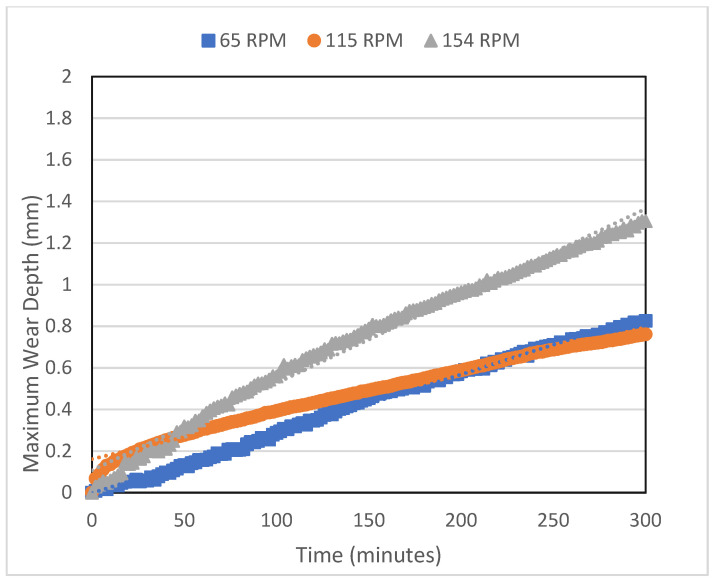
The maximum wear depth tested at 700 N and at different speeds (65 rpm, 115 rpm, and 154 rpm) versus time (minutes).

**Figure 10 polymers-16-00785-f010:**
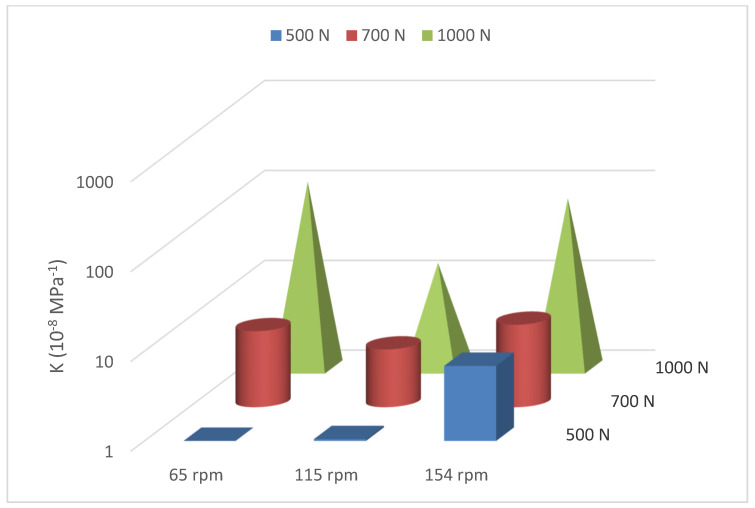
A 3D illustration of the average specific casing wear rates as a function of rotational speed and applied load.

**Figure 11 polymers-16-00785-f011:**
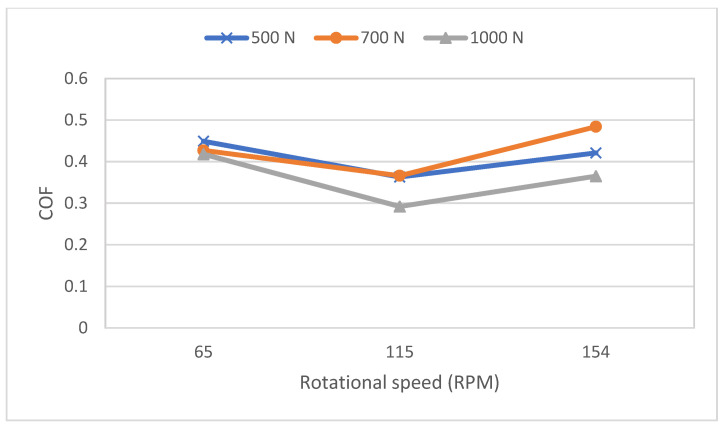
Average coefficient of friction of the casing as a function of the rotational speed and the applied load.

**Figure 12 polymers-16-00785-f012:**
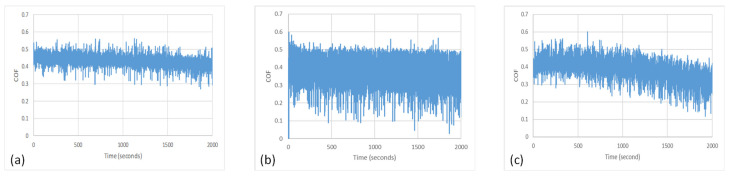
The values of the COF recorded in the first 2000 s of the experiment at 500 N and at (**a**) 65 rpm, (**b**) 115 rpm, and (**c**) 154 rpm.

**Figure 13 polymers-16-00785-f013:**
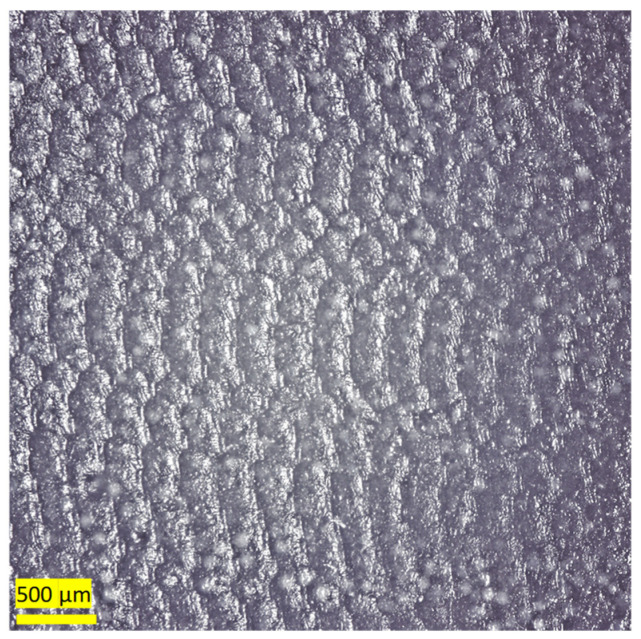
Optical microscope images for as-received sample surface.

**Figure 14 polymers-16-00785-f014:**
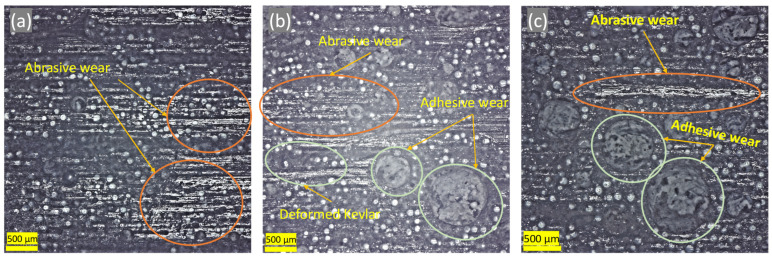
Optical microscope images of samples tested at (**a**) 65 rpm and 500 N, (**b**) 115 rpm and 500 N, and (**c**) 115 rpm and 700 N.

**Figure 15 polymers-16-00785-f015:**
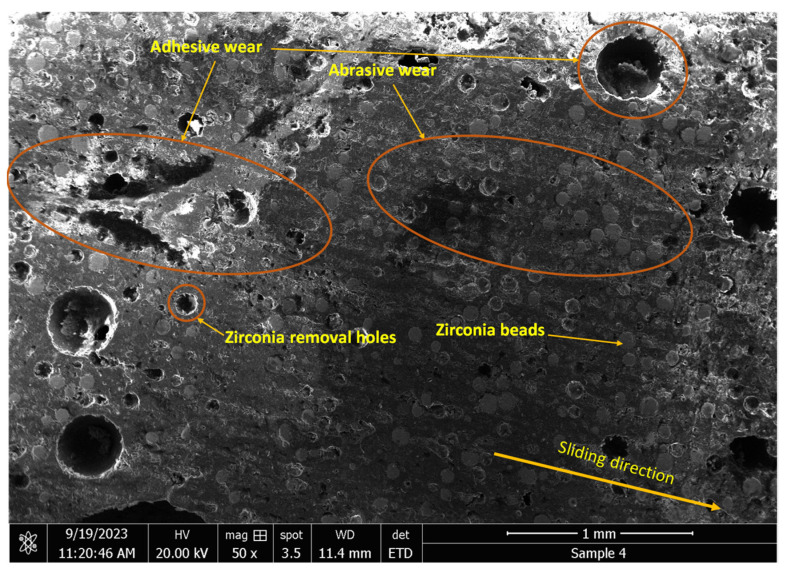
Electron image using SEM of the wear surface of the sample tested at 500 N and 115 rpm.

**Figure 16 polymers-16-00785-f016:**
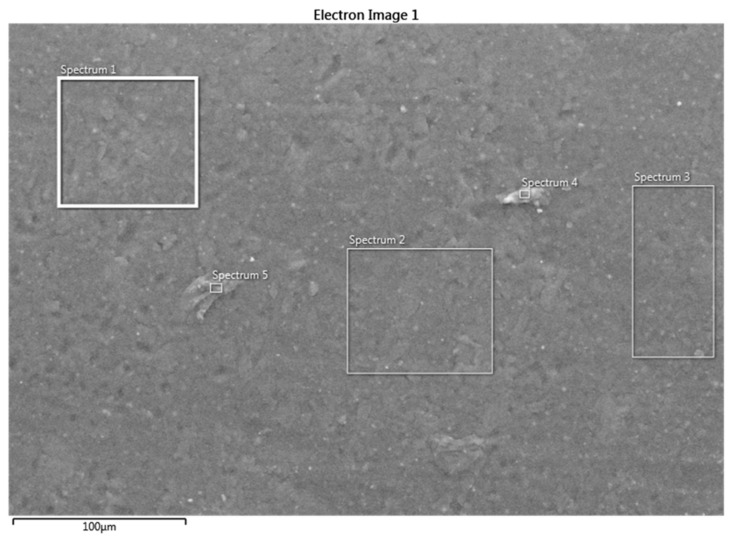
Electron image using SEM on an untested specimen.

**Figure 17 polymers-16-00785-f017:**
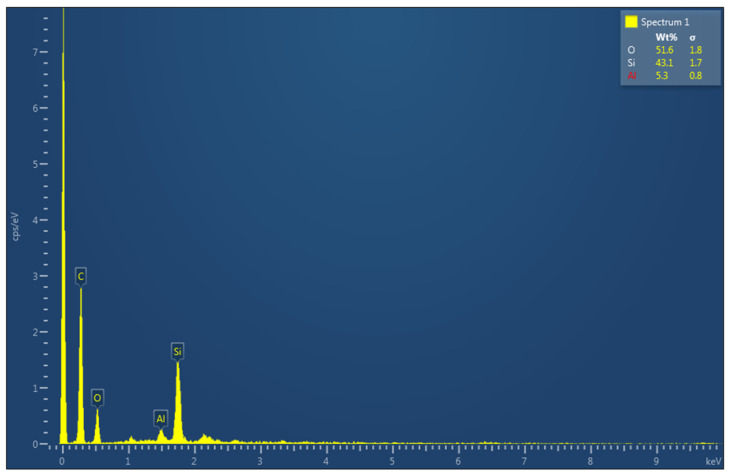
Spectrum 1 of the EDS analysis for untested specimens.

**Figure 18 polymers-16-00785-f018:**
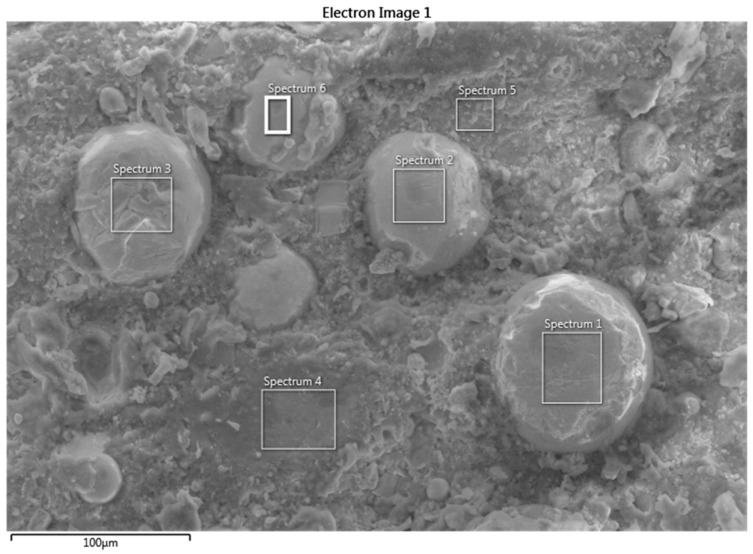
Electron image using SEM on a tested specimen at 500 N and 154 rpm.

**Table 1 polymers-16-00785-t001:** The dimensions (in mm) of the casing and DP-TJ samples.

	Outer Diameter	Inner Diameter	Sample Width	Composite Layer
Casing	244.5	211.5	35 ± 1	3–5
DP-TJ	137.2	78.0	146.0	35.6

**Table 2 polymers-16-00785-t002:** Mechanical properties of the casing as provided by the manufacturer.

Tensile Strength (MPa)	134.19
Compressive strength (MPa)	78.13
Bending Strength (MPa)	88.44
Shear Strength (MPa)	118.66
Shear Modulus (GPa)	3.2
Modulus of Elasticity (GPa)	6.8
Hardness	Shore D > 90

**Table 3 polymers-16-00785-t003:** Formulation of the water-based mud.

Water	As needed
Caustic Soda, ppb	0.5
Potato Starch, ppb	5
XC Polymer, ppb	0.75
Biocide (B-54 or equivalent), ppb	0.3
Marble Fine, pcf	Up to 67

**Table 4 polymers-16-00785-t004:** Test matrix.

Test No.	Speed (rpm)	Force (N)
1	65	500
2	65	500
3	65	700
4	65	700
5	65	1000
6	115	500
7	115	700
8	115	1000
9	154	500
10	154	500
11	154	700
12	154	1000

**Table 5 polymers-16-00785-t005:** Specific wear rate (K) and wear volume of Kevlar–epoxy–ceramic nanocomposite.

Test Number	Speed (rpm)	Side Load (N)	Total Wear Depth (mm)	Loss in DP-TJ Thickness (mm)	Wear Volume (mm^3^)	Casing Specific Wear Rate K × 10^−8^ (MPa^−1^)
1	65	500	0.180	0.05	70.04	0.929
2	65	500	0.103	0.06	31.33	0.276
3	65	700	0.826	0.15	683.95	8.62
4	65	700	0.637	0.151	474.76	5.43
5	65	1000	3.001	0.15	4525.7	115.16
6	115	500	0.339	0.15	186.9	1.046
7	115	700	0.760	0.15	621.76	4.398
8	115	1000	1.329	0.1	1599.95	14.439
9	154	500	0.792	0.125	647.86	5.125
10	154	500	1.140	0.2	1127.98	8.5683
11	154	700	1.305	0.15	1372.17	8.289
12	154	1000	3.137	0.24	4803	75.33

**Table 6 polymers-16-00785-t006:** EDS analyses on different spectra of untested samples.

Element	Spectrum 1 (wt.%)	Spectrum 2 (wt.%)	Spectrum 3 (wt.%)	Spectrum 4 (wt.%)	Spectrum 5 (wt.%)
O	51.6	54.5	54.4	57.8	65.2
Si	43.1	38.8	37.1	30.3	29.8
Al	5.3	6.7	8.5	11.9	5.0

**Table 7 polymers-16-00785-t007:** EDS analyses on different spectra of a sample tested at 500 N and 154 rpm.

Element	Spectrum 1 (wt.%)	Spectrum 2 (wt.%)	Spectrum 3 (wt.%)	Spectrum 4 (wt.%)	Spectrum 5 (wt.%)	Spectrum 6 (wt.%)
O	50	42.9	38.5	23.4	30.6	44.4
C	24.1	18.4	33.2	75.5	64.9	16.0
Zr	18.3	29.3	21.0	-	-	28.8
Si	5.7	7.6	5.9	1.1	3.1	8.5
Al	1.8	1.8	1.4	-	1.4	2.3

## Data Availability

All supporting data are provided in this paper.

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
