# Peer review of "Tribological Characterization of a Novel Ceramic–Epoxy–Kevlar Composite"

_polymers, 2024, doi:10.3390/polym16060785_

Round 1

Reviewer 1 Report

Comments and Suggestions for Authors

The article "Tribological Characterization of a Novel Ceramic-Epoxy-Kevlar-Composite" is interesting and requires minor review before publication. The authors examined commercially available materials, therefore justification of the relevance and significance of the research conducted is required.

Comments:

1. The introduction should emphasize the novelty of the research. To the extent that the analyzed material is new, the importance of the research in relation to the literature needs to be more broadly emphasized.

2.Experimental Procedure: On what basis did the authors select measurement parameters such as rotational speeds and side loads, do they correspond to operational conditions?

3. Table 4. Test matrix: What do the different abbreviations S1 to S9 mean?Is it the same material all the time and only the measurement parameters change? It is not clear, please add a sentence of explanation.

4. 4. Wear Mechanisms: This chapter is interesting, but the discussion of the results is too poor and requires supplementation with references to other research works.

5. Conclusions:

There are no conclusions emphasizing the essence of the research conducted. Practical aspects should be more specific to determine the scope of work of given materials.

Comments on the Quality of English Language

Minor editing of English language required.

Reviewer 2 Report

Comments and Suggestions for Authors

The authors reported the effect of side load and rotational speed on the tribological behavior of a novel ceramic-epoxy composite in Kevlar matrix. The result of the friction coefficient and the wear rate were collected. Moreover, the worn morphologies of the composites were provided. It’s an interesting work, however, the work is not meticulous enough and lack sufficient in-depth analysis.

Comments are as follows:

1.     Figures 5 and 6 show the positions of maximum and minimum hardness, nevertheless, the reason of distinct hardness caused by speeds and loads does not be given.

2.     The author believed that the lubricant layer played a role in the tribological performance, please provided the characterization about the transfer film.

3.     Figure 13a is the untested sample, however, there seems to be a lot of fatigue crack on the untested sample surface.

4.     Please provide the SEM images of the worn surface and give the detail explanation about the worn mechanism. The XPS or XRD of the worn surface can also be given to illustrate the tribological behavior during the sliding process.

Comments on the Quality of English Language

Extensive editing of English language required.

Round 2

Reviewer 2 Report

Comments and Suggestions for Authors

The work can be accepted.